# Acceptance of Muffins (Sweet and Savory) with the Addition of *T. molitor, A. diaperinus, A. domesticus, R. differens*, Considering Psychological Factors (Food Neophobia Scale, Consumer Attitude)

**DOI:** 10.3390/foods13111735

**Published:** 2024-06-01

**Authors:** Aleksandra Mazurek, Agnieszka Palka, Stanisław Kowalski, Magdalena Skotnicka

**Affiliations:** 1Department of Commodity Science, Faculty of Health Sciences with the Institute of Maritime and Tropical Medicine, Medical University of Gdansk, 80-210 Gdansk, Poland; aleksandra.mazurek@gumed.edu.pl; 2Department of Quality Management, Faculty of Management and Quality Science, Gdynia Maritime University, 81-225 Gdynia, Poland; a.palka@wznj.umg.edu.pl; 3Department of Carbohydrate Technology and Cereal Processing, Faculty of Food Technology, University of Agriculture in Krakow, Al. Mickiewicza 21, 31-120 Krakow, Poland; rrkowals@cyf-kr.edu.pl

**Keywords:** edible insects, acceptability, consumer attitudes, food neophobia scale, buffalo worm, mealworm, house cricket, grasshopper

## Abstract

The aim of the study was to analyze the acceptance of muffins containing a 15% addition of powder from four edible insect species (*Alphitobius diaperinus, Tenebrio molitor, Acheta domesticus, Ruspolia differens*) in both savory and sweet versions, focusing on the psychological factors influencing their consumption. The study involved 106 adult consumers. Initially, the level of food neophobia (FNS) among participants was determined. Over 80% displayed low to medium levels of neophobia. Similar results were obtained when assessing attitudes towards insects, with most participants showing positive and ambivalent attitudes. Based on these findings, the acceptance of insect-based muffins was evaluated. The level of acceptance of insects varied and depended mainly on taste, smell, and texture. Participants with lower levels of neophobia and positive attitudes towards consuming insects generally rated the insect muffins higher compared to those with higher levels of neophobia and negative attitudes. The sweet versions of insect powder muffins were rated higher, which also indicates preferences and dietary habits. Products with grasshopper powder (GS, GCL) were rated the lowest for both taste versions. Conversely, products based on buffalo worms (BS, BCL) were seen as having the greatest potential for acceptance. Understanding consumer attitudes, neophobia, and levels of acceptance provides valuable insights for designing new insect-based foods.

## 1. Introduction

Rising incomes and urbanization are causing a global shift in dietary habits, with traditional plant-based diets being replaced by meat-rich diets, particularly those high in animal protein [1,2]. In highly developed countries, meat is considered a key component of their diet, leading to an increase in animal production [3]. In some regions of the world, such as parts of Africa and South Asia, access to animal protein is limited due to high costs, resource scarcity, or a lack of infrastructure. On the other hand, animal protein production has a significant environmental impact [4,5,6]. Livestock farming requires large amounts of land, water, and feed, leading to deforestation, land degradation, and water pollution. Additionally, greenhouse gases emitted by the agricultural sector, mainly methane from cattle breeding, are a significant contributor to global warming [7,8,9]. Considering all the burdens associated with traditional farming and nutrition in highly developed countries, as well as securing adequate energy in diets in countries at risk of malnutrition, there is growing interest in alternative ways of obtaining protein, such as the use of edible insects in production [10,11,12].

Edible insects are a source of protein that could serve as an alternative to livestock meat [13]. The introduction of edible insects for widespread consumption is one of the methods to ensure food security on a global scale, considering the growing population and the high environmental costs of obtaining animal protein [14,15]. By introducing substitute sources of protein, such as insects, environmentally harmful factors can be partially reduced. Insects have a highly efficient metabolism, a short generation time, contain comparable amounts of high digestibility compared to conventional sources, and have a lower impact on the generation of greenhouse gases [9,16,17,18,19,20].

From a nutritional point of view, insects not only provide energy, but in addition to protein, they are a good source of fiber, vitamins, minerals, and bioactive substances [11,21,22,23]. Additionally, insects provide fiber in the form of chitin, as well as omega-3 and omega-6 fatty acids. It has been shown that the consumption of insect powder can affect the composition of the microbiota in the colon [24]. Thus, food enriched with edible insects represents a new type of functional food with significant development potential [25,26]. Despite the many environmental, sustainable agriculture, and health benefits that can result from consuming insects, they still do not enjoy widespread popularity.

One of the greatest challenges in introducing insects into the diet in Western cultures is overcoming the cultural and psychological barriers associated with consuming insects. In European countries, where insect consumption is not a tradition, it may require significant educational and marketing efforts to change the perception of insects from “disgusting” to “tasty and healthy”. Acceptance is a prerequisite for introducing new or modified food products for widespread consumption [27,28,29,30,31]. In the case of edible insects, the main barrier that prevents consumers from incorporating insects into their diet is their psychological effects. These include food neophobia, disgust, associations with dirt and poverty, and the perception of products as unsuitable for enrichment with insects [32]. Regarding entomophagy, food neophobia is particularly high in Western culture [33]. For this reason, acceptance studies should be expanded to include comprehensive research on opinions, attitudes, beliefs, and nutritional knowledge. Current data on insect consumption indicate that consumers perceive only certain food products as suitable for enrichment with insects, depending on whether the product is savory or sweet and its form [34]. Moreover, Orsi et al. [35] suggest that enriching processed food with insect powder rather than whole insects is the most promising strategy for implementing entomophagy in European countries.

In countries where insect consumption is widespread, the motivation for their consumption is based on factors related to sensory characteristics, availability, affordability, and preference. In Western countries, the key determinants of their consumption are nutritional value, sustainability, health benefits, consumers, product knowledge, their previous experience, culture, neophobia, and disgust [36]. However, the most important factors include overall acceptance and willingness to consume a given food [37,38,39]. Authors of previous studies suggest gradually introducing insects into the diet. One way to do this is by enriching traditional foods with insect powder [40,41,42]. Among many proposals, cereal-based products with insects have the greatest market and sensory potential. They are also the most numerous on the market and are highly rated by consumers [43,44,45]. Taking into account those facts and the current state of knowledge, the aim of this study was to analyze the acceptance of muffins with a 15% addition of four species of edible insects (*Alphitobius diaperinus*, *Tenebrio molitor*, *Acheta domesticus* and *Ruspolia differens*) in salty and sweet versions compared to the control sample and to analyze attitudes and psychological factors related to neophobia among the surveyed consumers.

## 2. Materials and Methods

### 2.1. Assessment of the Level of Food Neophobia (FNS)

The level of food neophobia was measured using the Food Neophobia Scale (FNS) by Pliner and Hobden [46]. The FNS scale helps determine whether a consumer’s eating behavior is more oriented towards the desire to discover new food products (neophilia) or to remain solely with known and accepted products. Moreover, the Food Neophobia Scale is a widely used and validated tool [47]. The FNS questionnaire included the following statements:I am constantly trying new dishes (R).I do not trust new food.If I don’t know what’s in the food, I won’t try it.I like dishes from different countries (R).Ethnic food looks too strange to eat.At parties, I try new food (R).I am afraid to eat things I have never eaten before.I am very careful about the food I will eat.I will eat almost anything (R).I like trying new ethnic restaurants (R).

The items marked with (R) are reversed scored, meaning that agreement with these statements indicates lower food neophobia or a tendency toward neophilia.

To determine the reliability of the questionnaire, the Cronbach’s alpha coefficient was set at 0.85. This result demonstrated good internal consistency, as the value exceeded 0.7 [48]. Respondents rated the extent to which they agreed with the statements in the questionnaire using a 5-point Likert scale, from 1—“Strongly disagree” to 5—“Strongly agree”. For statements marked with (R), reverse scoring was applied [49]. The scores obtained ranged from 10 to 50. Based on the results of the FNS (Food Neophobia Scale) questionnaire, the study group was divided into those with low, medium, and high levels of neophobia according to the following criteria [50]:-Low level of food neophobia—FNS score < mean value minus SD,-Medium level of food neophobia—FNS score from mean value minus SD to mean value plus SD,-High level of food neophobia—FNS score > mean value plus SD.

The questions from the questionnaire are fully adapted from the FNS scale, while the division into three levels at such intervals is also consistent with the method described above.

### 2.2. Assessment of Consumer Attitudes

The level of neophobia and attitudes towards entomophagy are significant factors determining the acceptance of insects as food [51,52]. The study group was divided into individuals with positive, ambivalent, and negative attitudes, like those commonly used in attitude research [53], and the overall acceptance of muffins was compared across these groups. The division criteria were identical to those used for the FNS (Food Neophobia Scale), incorporating the mean and standard deviation. The attitude questionnaire contained 8 statements rated on a 5-point Likert scale, from 1—“strongly disagree” to 5—“strongly agree”. A higher score indicated a more positive attitude towards consuming insects. The possible scores ranged from 8 to 40. The author’s self-developed attitude questionnaire included the following statements:I believe that consuming insects has health benefits.I believe that eating insects supports environmental protection.I think insects are suitable as an addition to any type of food.If insects were tasty and affordable for me, I would be willing to include them in my diet.I believe that eating insects is a good source of protein.I have previously been interested in trying edible insects.I believe that eating insects is dangerous for your health (R).Insects make me think of dirt (R).

For statements 7 and 8 marked with (R), reverse scoring was applied [54]. The Cronbach’s alpha coefficient was 0.74. The same division criteria used for the FNS (Food Neophobia Scale) results were applied to the attitude questionnaire:—Negative attitude—attitude score < mean value minus SD,—Ambivalent attitude—attitude score from mean value minus SD to mean value plus SD,—Positive attitude—attitude score > mean value plus SD.

### 2.3. Acceptability of Insect-Based Muffins

Volunteers for the study were selected from the ConsumTest 2.0 database at the Medical University of Gdańsk. Before participating in the study, participants signed a consent form and were informed about the course of the study. Initially, 140 individuals registered. Due to tobacco smoking, 10 people were excluded, and 19 people withdrew during the study. Due to incomplete information in the questionnaires, 5 people were excluded. Ultimately, 106 individuals aged 20–27 years, who declared no allergies or chronic diseases, were not taking medications regularly, and were not following elimination diets completed the study, as shown in Figure 1. The study was approved by the Bioethics Committee for Scientific Research at the Medical University of Gdansk (No. KB/336/2023).

The muffins were served to participants one day after baking. On one day, five variants of sweet muffins were evaluated (control sample and muffins with the addition of four types of insects). Two days later, five variants of savory muffins were assessed. Participants rated the muffins on an unstructured visual scale from 0—“Completely unsatisfactory” to 10—“Completely satisfactory” in terms of appearance, smell, taste, texture, and overall acceptance [55]. The following descriptors were adopted: appearance (0 (completely unsatisfactory) –10 (completely satisfactory)), taste (0 (very unpleasant)–10 (very tasty)), texture (0 (completely unsatisfactory for me) –10 (completely satisfactory for me)), and overall acceptance regarding all analyzed descriptors (0 (very unsatisfactory)–10 (very satisfactory)). The selection of descriptors included in the sensory evaluation was based on PN-EN ISO 5492:2009 “Sensory analysis—Terminology” [56] and PN-ISO 11035:1999 “Sensory analysis—Identification and selection of descriptors for determining the sensory profile using multidimensional methods” [57].

Names and definitions of descriptors:

Appearance: The overall visual impression that a product creates, consisting of a series of individual visually perceptible features (e.g., shape, color, gloss).

Aroma: The sensation perceived by the sense of smell.

Taste: The sensation perceived by taste receptors on the surface of the tongue when stimulated by a stimulus.

Texture: The intensity of sound associated with the deformation of the sample (e.g., during biting).

Acceptance: The overall sensory impressions of the consumer towards the presented sample.

### 2.4. Muffin Preparation

Four species of insects were used to prepare the muffins: mealworm (*Tenebrio molitor)*, buffalo worm (*Alphitobius diaperinus)*, house cricket (*Acheta domesticus)*, and grasshopper (*Ruspolia differens)*. *A. diaperinus* and *T. molitor* were in larval form, while *A. domesticus* and *R. differens* were in adult form. The lyophilized insects came from Insecten kwekrij van de Vn Fortweg, Deurne, Netherlands. The whole insects were ground into powder using a laboratory mill (IKA, A11 basic, Staufen, Germany). The muffins were made with the following ingredients: wheat flour (type 500), milk (2% fat), canola oil, eggs, salt, sugar, and baking powder, which were purchased at a local market. Wet and dry ingredients were combined separately and then mixed until a uniform consistency was achieved. The batter was poured into muffin molds and baked at 180 °C for 25 min in a preheated oven. The muffins were then left to cool at room temperature.

The modifications involved replacing 15% of the wheat flour in the basic recipe with insect powder. The control sample was prepared without the addition of insect powder but with the addition of wheat bran and a higher content of oil. The salt and sugar content was established using the ISO 3972:2011 [58] standard “Sensory analysis-Methodology-Method of investigating sensitivity of taste”. Two versions were prepared for analysis: savory and sweet. Ultimately, 10 test samples were used to assess acceptance: 5 savory versions along with the control sample (OCL-control; MCL-mealworms salty; BCL-buffalo worms salty; CCL-house cricket salty; GCL-grasshopper salty) and 5 sweet versions against the control (OS-control; MS-mealworms sweet; BS-buffalo worms sweet; CS-house cricket sweet; GS-grasshopper sweet). The composition of the muffin recipes is presented in Table 1.

### 2.5. Nutrient Composition of Muffins Enriched with Insect Meals

The following parameters of the edible insect powder were analyzed: the ash content (AOAC 923.03); protein content (AOAC 950.36) (the protein content was calculated applying a conversion factor of 6.25); crude fat content (AOAC 935.38); water content (AOAC 925.10); and the total, soluble, and insoluble dietary fiber content (AOAC 991.43). [59]. The determination of the chitin content was adapted based on references by Tsurkan et al. [60]. The analyses were performed in triplicate.

### 2.6. Statistical Analysis

Statistical analysis was carried out using Statistica 13.3 (StatSoft, Krakow, Poland). In the statistical calculations, the Shapiro-Wilk test was utilized to analyze the normality of the distribution of variables at a significance level of *p* ≤ 0.05. Results are presented as the mean ± standard deviation. To identify differences and relationships between the sweet and savory versions, the Wilcoxon test was used at α = 0.05. Furthermore, the Kruskal-Wallis test was applied to reveal relationships between attitudes, levels of neophobia, and acceptance of selected insect powder versions at a significance level of α = 0.05. Primary data obtained from the consumer evaluation study were subjected to statistical calculations, based on which a regression analysis was conducted for each of the tested insect powder variants. Multiple regression was used to determine relationships between several independent variables, such as appearance, smell, texture, and taste, and the dependent variable, which is acceptance.

## 3. Results

In recent years, there has been a significant increase in interest in alternative protein sources, focusing consumer and research attention on entomophagy, or the consumption of insects. Despite potential health and environmental benefits, overcoming cultural and psychological barriers remains a challenge that can affect the acceptance of this form of food, especially in Western countries. A key aspect of understanding these challenges is determining the level of food neophobia, which is the resistance to trying new food products, and analyzing consumer attitudes towards insects as food.

### 3.1. Food Neophobia Scale

In the case of the Food Neophobia Scale (FNS), after reversing the coding of five neophilic statements, the average neophobia score was calculated for each of the ten factors. This value potentially ranged from 10 to 50, where a higher score represented a higher level of food neophobia. A graphical representation of the group’s characteristics is presented in Figure 2. Individuals classified into the neophilic group (with a low level of food neophobia), who scored between 10 and 16 on the FNS (*n* = 24), accounting for 22.64% of the entire study group, exhibit a range of psychological and behavioral traits reflecting their openness to new culinary experiences, including trying ethnic dishes. Their approach to food is curious and exploratory, and they are less concerned about the potential negative consequences of consuming unknown food products, which may be associated with a higher level of trust in new dishes and food safety regulations.

Individuals with scores ranging from 17 to 29 on the FNS were classified into a group with a medium level of neophobia, comprising 65 people or 61.32% of the study group. Individuals with a medium level of food neophobia occupy an intermediate position between openness to new culinary experiences and conservatism in their dietary preferences. Although they may be open to new food trends, they may simultaneously exhibit resistance to extreme novelties, such as the consumption of insects, which are significantly different from their daily diet.

Those who scored between 30 and 50 on the FNS were classified into the high neophobia group (*n* = 17), corresponding to 16.04% of the total. These individuals show a high degree of conservatism or even aversion towards new foods. They tend to stick to well-known and tested products, avoiding novelties in their diet. They show significant resistance to changes in their dietary habits, which may be motivated by fear of the unknown consequences of consuming a new type of food. The only common feature across all groups was a concern for the food they consume. This statement did not differentiate the groups. Comparing the results of the response to question 8, more than 70% in all three groups agreed with the statement, indicating that the quality of the food consumed is very important to the participants.

### 3.2. Assessment of Attitudes

The study also included an analysis of attitudes towards insects as food, where results revealed differences in perceptions of health safety, health benefits, environmental benefits, and the necessity of enriching various types of food with insects. Figure 3 presents the results of the attitude questionnaire, which classified respondents into three categories of attitudes: negative, ambivalent, and positive.

Individuals classified with negative attitudes, scoring between 8 and 22 points on the attitude questionnaire (*n* = 21), which accounted for 19.81% of the study group, demonstrated a negative stance towards the emotional components of attitudes—associating insects with dirt and deeming them an inappropriate addition to all types of food. The majority of this group would not be willing to include insects in their diet, even if they were tasty and inexpensive. The attitudes of this group were ambivalent regarding issues such as health safety and the potential health and environmental benefits of consuming insects. Among the participants in this group, 43% agreed with the statement that insects are a good source of protein.

The largest group, 61.32%, similar to the analysis of food neophobia (FNS), consisted of those with ambivalent attitudes. These were individuals who scored between 23 and 32 points (*n* = 65). Over half of the group with ambivalent attitudes perceived insects as a good protein source and a factor conducive to environmental protection. Additionally, they considered entomophagy safe from a health perspective, though opinions were divided regarding potential health benefits. A similar situation occurred regarding the suitability of enriching all types of food with insects and the previous willingness to consume them. More than half of this group expressed no associations between insects and dirt and were willing to incorporate them into their diet. Individuals classified as having positive attitudes scored between 33 and 40 on the attitude questionnaire (*n* = 20). All participants in this group agreed that consuming insects is safe for health, constitutes a good protein source, and aids environmental protection. Most of this group (75%) had previously been willing to try edible insects. Entomophagy was viewed as a health-promoting phenomenon by 80% of the group. Half of them considered insects an appropriate addition to all types of food, while the other half were undecided on this issue. All declared their willingness to include insects in their diet, provided they were tasty and reasonably priced.

### 3.3. Nutrient Composition of Muffins Enriched with Insect Meals

The analysis of the composition of insect powder indicated that the house cricket (*A. domesticus)* contained the highest protein content. Simultaneously, it was also the richest in fat, which could have influenced the sensory evaluation of all quality descriptors, as fat content shapes and enhances flavor, aroma, and rheological impressions. The house cricket and grasshopper had the highest dietary fiber content. This is related to the form of the insect used for the studies, as the flour from these insects was powdered from the mature imago form, unlike the other two, which were freeze-dried as larvae. In all the analyzed insects, the majority was insoluble fibers along with chitin. Only in two—*A. domesticus* and *T. molitor* were small amounts of water-soluble fiber observed, which likely did not affect the differences in taste perception, aroma, structure formation, and overall appearance depending on the addition of powder to the test sample in Table 2.

### 3.4. Acceptability of Insect-Based Muffins

Among the savory and sweet variants, muffins with *R. differens* (GCL, GS) received the lowest ratings for all tested characteristics: appearance, smell, taste, texture, and overall acceptance compared to other additions. Sweet muffins with grasshoppers (GS) scored better in terms of appearance, texture, and overall acceptance than their savory counterparts (GCL). The exception was the taste rating, where both variants received scores that did not differ statistically significantly. Values marked with the same symbols within a group exhibited similar values within the tested sensory characteristic. The results of the sensory evaluation of the muffins in both the salty and sweet versions are presented in Table 3.

The control sample (OCL, OS) was rated the highest in both sweet and salty versions. All test variants with the addition of insect powder were rated lower. The acceptance of the sweet control sample was significantly higher than MS *p* = 0.0234, BS *p* = 0.0232, CS *p* = 0.0056, and GS *p* = 0.1982. Simultaneously, the savory control sample was significantly higher than the test samples (MCL *p* = 0.1667; BCL *p* = 0.0223; CCL *p* = 0.0035; GCL *p* = 0.0092 at *p* = 0.05). The sweet version was rated slightly higher than the savory option. The analysis of the Shapiro-Wilk test *p* = 0.0565, at the adopted significance level, indicates no basis for rejecting the hypothesis assuming the normality of the variable distribution. Therefore, to examine the relationship between the sweet and salty versions, the Wilcoxon test (*p* = 0.0077) was conducted, indicating that there is a statistically significant difference between the acceptability of insect muffins in the sweet and savory versions. The differences in acceptability of the individual variants are presented in Table 4.

For muffins with the addition of mealworm and grasshopper, statistically significant differences were found between the sweet and savory versions. In both cases, the acceptance for the sweet version was significantly higher than for the savory version. This indicates acceptance is determined not only by the insect addition but also by the recipe and composition, which are important cues for producers and the food market.

Among the analyzed attributes, appearance was rated the highest, which, with the exception of muffins with the addition of grasshopper, consistently exceeded 8.6 points for the rest of the test samples regardless of the flavor version. Considering the smell, it was rated relatively high. Apart from the control samples (OCL and OS), the highest aroma ratings were given to *A. diaperinus* and *A. domesticus* in the savory version, while in the sweet version, all test variants received high scores above 8.2 points.

The most significant differences were found in the taste evaluation. Regardless of the proposed version, all samples enriched with insect powder received significantly lower ratings. The highest taste acceptance was declared for buffalo worms, both for the salty variant at 7.88 and the sweet variant at 7.71. Conversely, the savory variant with the addition of grasshopper powder was rated the lowest at 5.49. The similar situation was found for the sweet version at 5.55. Regardless of the recipe and composition, grasshoppers obtained the lowest levels of taste acceptance. Texture was the last attribute contributing to overall acceptance, which did not have a significant impact on final acceptance by consumers. The structure of the muffins in all possible tests exceeded the level of 7.21, suggesting that the texture of the cupcakes with added insect powder was appropriate and acceptable.

Despite relatively high ratings for individual attributes, overall acceptability was decidedly lower for all 10 samples. This is very intriguing; however, it should be emphasized that the overall trend and the ranking from the lowest to the highest-rated test sample were maintained. The proposed variants indicated that the sweet version was better accepted by the group of participants in the study. Likely, taste was decisive, which had the strongest impact on the level of acceptance.

A multiple regression analysis was employed to estimate the relationship between the acceptance of selected descriptors of sensory quality of muffins with added insect powder and the acceptance of these products as perceived through the prism of their overall quality rating. The dependent variable was the acceptance of muffins, while the independent input variables were the acceptance ratings of individual sensory quality parameters: appearance, smell, taste, and texture.

During the analytical procedure, independent variables were narrowed down to critical parameters. The significance of the generated acceptance models was assessed at a significance level of *p* ≤ 0.05. Multiple regression analysis was used for all tested muffin variants in sweet and savory versions to estimate the impact of individual variables on the overall level of acceptance. Table 5 presents the obtained regression equations for overall acceptability. The multiple regression analysis showed that for the tested muffins with insect powder, the smell was insignificant and did not have a significant impact on the overall acceptability of the tested samples, or its role was minimal. In summary, the obtained results indicate that the level of acceptability resulted from predictors contained in the regression equation, with taste, appearance, and texture having the greatest importance for overall acceptability.

The coefficient of determination R^2^, which ranges from 0.61 to 0.85, shows that the regression models explain the variability in the acceptance ratings of the muffins well. This indicates the high precision of the models in predicting consumer preferences based on the evaluated sensory characteristics.

The analysis of the equation suggests that taste is the key determinant of acceptance. The significance of this factor in the sweet version, where taste has a decidedly greater impact, confirms that consumers are primarily guided by taste sensations when evaluating food products containing unusual ingredients such as insect powder. Appearance also plays an important role, particularly in the context of the first impression, which can influence the willingness to try a new product. Meanwhile, the impact of texture was variable, depending on the test version. Texture proved to be more significant in the sweet trials. Regression equations indicate the diverse influences of sensory factors on acceptance. For most variants, texture and taste turned out to be the most influential variables. For example, in the case of muffins with the addition of *R. differens* (GS) and *A. domesticus* (CS) powder, taste had the greatest impact (coefficients of 0.69 and 0.64 for the sweet version, respectively). In the savory version, the importance values were similar, also indicating that taste, but also texture, was the deciding predictor shaping overall acceptance. This is extremely important because technological conditions associated with the addition of insects can effectively lower acceptance regardless of other quality characteristics, hence the importance of selecting the right recipe and proportion of the mixture for baking. Although smell is usually an important element in food evaluation, in the case of insect muffins, it had limited significance. This may be due to the fact that the characteristic smell of insect powder was not dominant or was well masked by other ingredients in the muffins.

### 3.5. Relationship between FNS, Attitudes, and Level of Acceptance

The next step of the experiment was to examine the relationship between the level of designated neophobia and the consumers’ assessment of acceptance of muffins with the addition of insect powder in both sweet and savory versions. For this purpose, a Shapiro-Wilk test for normality of the variable distribution was conducted for the sweet version with *p* = 0.0046 for low neophobia, *p* = 0.0040 for medium neophobia, and *p* = 0.2300 for high neophobia. Only in the case of high neophobia there was no basis reject the hypothesis assuming the normality of the acceptance variable distribution. For the savory version, for high neophobia *p* = 0.0009 and medium *p* = 0.0016 neophobia levels, the Shapiro-Wilk test for *p* = 0.05 indicated a rejection of the hypothesis assuming the normality of the variable distribution. Only for low neophobia *p* = 0.0864, which was higher than α = 0.05, it was concluded that there was no basis for rejecting the hypothesis assuming the normality of the variable distribution (acceptance).

The results of the Kruskal-Wallis test for the sweet version *p* = 0.0067 and the savory version *p* = 0.0047, at the adopted significance level of α = 0.05, indicate that there is a statistically significant difference in the overall acceptance rating of muffins with insects prepared sweetly and salty. In both cases, acceptability significantly decreased with an increase in the level of neophobia Figure 4. Culinary neophobia, or the fear of trying new foods, had a particularly strong impact on the evaluation of products containing insects, which is understandable considering the unusual ingredient that insects represent [61]. The lowest acceptance ratings were recorded among individuals who exhibited the highest level of neophobia. These results are particularly interesting as they suggest that psychological barriers, rather than taste or other sensory characteristics of the product, may play a key role in limiting the willingness to incorporate insects into the diet.

In the study evaluating the acceptance of products containing insects as an ingredient, respondents’ attitudes were classified into three main categories: positive, ambivalent, and negative. This classification aimed to deepen the understanding of how attitudes toward insects as food could influence the readiness to consume them. The analysis of the normality of the data distribution for acceptance of insects in sweet products showed differences depending on attitudes. For the group with negative attitudes, the *p*-value was 0.2388, indicating no basis to reject the hypothesis of normality distribution at the significance level of α = 0.05. However, for ambivalent and positive attitudes, the *p*-values were 0.0084 and 0.0079, respectively, suggesting that the distributions for these groups are not normal.

These differences constitute an interesting point of analysis, as they suggest that people with different attitudes towards consuming insects may evaluate these products differently. This can affect the results of taste and sensory tests and thus, the overall market acceptance. In the case of savory products, the Shapiro-Wilk test for all attitude groups indicated a rejection of the hypothesis about the normality of the variable distribution, with *p*-values of 0.0095 for negative attitudes, 0.0048 for ambivalent attitudes, and 0.0011 for positive attitudes. These results suggest that regardless of attitude, the perception of salty insect products significantly deviates from the expected norm, which can influence how they are evaluated by different groups.

Comparable to the neophobia analysis, the results of the Kruskal-Wallis test (*p* = 0.0003 for sweet; *p* = 0.0015 for salty) suggest that there is a statistically significant difference in the overall acceptance of insects prepared both sweetly and savory. When analyzing this data, it is also important to understand how acceptance data relate to specific product characteristics, such as the type of insect added or the flavor version. Figure 5 in the study illustrates the trend where acceptance of insect-containing products generally increases depending on attitudes, which may be a result of taste adaptation or increasing consumer openness to new culinary experiences. In summary, understanding attitudes towards insects, assessing the level of neophobia, and their impact on product acceptance is crucial for developing marketing and product strategies that can enhance the acceptance of such food innovations among different consumer segments.

## 4. Discussion

The acceptance of insects as a food ingredient in products such as muffins and other sweets and snacks depends on many factors, including the level of food neophobia and attitudes towards new foods. Research results from various regions and communities show significant differences in the degree of acceptance of insects, which can arise from both cultural and individual conditions [37,39]. Understanding the level of neophobia and consumer attitudes towards insect-based food is crucial and allows for predicting how consumers will react to the introduction of insect products into the market. Individuals with low levels of neophobia are more inclined to try such products, which can accelerate their acceptance. Our research results confirm this. In all cases, acceptance among this group was the highest. Similar studies have been conducted by Erhard et al., Shelomi, and Sogari et al. [61,62,63], indicating that individuals who are strongly neophobic may remain reluctant to incorporate insects into their diet, even if they are informed about the health, environmental, and economic benefits. In many publications, even a high percentage of neophobia does not limit the possibilities of introducing insects into the daily diet. Research conducted by Caparros Megido et al., Phonthanukitithaworn et al., and Ros-Baró et al. [2,64,65] shows that the possibilities for edible insects to become a common food ingredient in Western European populations are relatively high and mainly dependent on targeted marketing strategies. This aligns with our results, where consumers are also willing to buy and cook insects at home if they can associate them with familiar tastes and preferred dishes. Knowledge of the level of neophobia can shape educational and marketing strategies in the future that help overcome psychological and cultural barriers. Tailoring the right message to the specific concerns and expectations of different consumer groups is key in the process of promoting a new category of food based on edible insects [43,66,67]. The authors suggest using the Persuasion Knowledge Model as a subliminal message in appropriately constructed advertising. Utilizing psychological mediators, such as communicating emotions and a sense of trust in the advertisement, can lead to a positive perception of the product, such as edible insects [68]. Research conducted by Park and Cho demonstrated an effective reduction in the level of neophobia and an increase in the level of new food acceptance using a taste education program modified to indicate connections with elements of the native food culture [69]. The education program used in Park and Cho’s research seems promising in the context of entomophagy promotion.

In addition to determining the level of neophobia, an important determinant for understanding the motivation and attitudes towards consuming insects is consumer attitudes. Evaluating these attitudes appears to better predict intentions regarding both direct and indirect entomophagy [70]. Individuals with positive attitudes are more open to trying insects and often perceive them as an alternative, sustainable source of protein, as reflected in our studies. A study conducted by Tuccillo et al. [71] among Italian consumers found that 41% of respondents had a positive attitude towards entomophagy, while 27% exhibited negative attitudes. An ambivalent stance was shown by 32% of respondents. In our study, the largest group consisted of consumers with ambivalent attitudes, comprising 61.3% of the participants, representing more than half of the subjects. Our findings align with the hypothesis proposed by Florença et al. and Shiv and Fedorikhin [29,72], who suggest that unfamiliar foods often elicit ambivalent attitudes, which include both positive and negative elements. Such a distribution of attitudes, as in our study, is favorable but also poses a challenge. Over 80% of respondents demonstrate positive or ambivalent attitudes, offering opportunities to persuade undecided consumers and to develop the market for such foods with patience and consistent efforts.

In conclusion, understanding the level of neophobia and consumer attitudes towards insect-based foods is essential to creating effective marketing, educational, and product strategies aimed at introducing and raising awareness about these products in markets with diverse cultures and dietary traditions. In recent years, there has been an increasing interest in insect consumption in countries where insects were not traditionally consumed. Currently, the insect food market encounters many difficulties related to the acceptance of such foods. Influential factors include a tangible fear of trying new foods. There is a lack of adequately presented knowledge about the health, environmental, and economic benefits associated with consuming insects. Barriers also include somewhat unclear legal regulations concerning the breeding, processing, and sale of insects. Many people reject insects as food due to fear of the unknown and uncertainties about the taste, texture, and safety of consuming insects [73,74]. Moreover, the aforementioned inhibiting factors, perhaps the most crucial appears to be the acceptance of insect-based products. It is essential to propose such products in such a way that even consumers who are reluctant to change can overcome their prejudices. One way to achieve this is by using insect powder, which can be more easily incorporated into existing recipes than whole insects [75,76]. Another aspect is the use of insects in appropriate proportions without compromising the basic quality characteristics of commonly liked products such as cereals, pasta, and sweets [61,77,78,79,80,81]. These products seem to have the greatest market potential. In our study, commonly liked muffins were used, enhanced with four different types of insects. The choice was dictated not only by their availability but also by their popularity in the market. According to Van Huis [17] crickets (*A. domesticus)* are the most widely accepted and consumed farmed insects worldwide, which was not confirmed in our findings. The best-rated insect was the buffalo worm [82]. This was also confirmed by the authors’ previous research [83]. Alongside the cricket and buffalo worm, the mealworm [84] also enjoys high acceptance. Consumers often pay attention to the form of the insect. Insects in larval form are rated significantly better [12]. They contain small amounts of chitin, which can affect the taste and texture of various food products containing insects. In our study, variants with the insect in the adult its form (imago) were rated significantly lower. The acceptance of muffins with the addition of grasshoppers, both in sweet and salty versions, was rated the lowest. This is not particularly surprising since grasshoppers are often rated very low in other publications. Here, the decisive factor is likely not the insect addition per se but rather the taste and aroma of this specific insect [85]. Although research by Ochieng et al. [86] indicates that volatile substances responsible for the aroma of cookies with *R. differens* are associated with a positive sensation, in this case, there was no comparative analysis with other insects.

Differences in the evaluation of acceptance may result from various factors, such as specific flavor and aroma compounds that arise from different perceptions of amino acids, proteins, and fats contained in individual edible insects and within a single species [87,88]. Insects vary in amino acid composition, which consequently affects the compounds formed during Maillard reactions during baking, thereby shaping taste impressions differently [89].

Additionally, the study proposed the evaluation of two flavor variants, sweet and salty. In all cases, the acceptance of the sweet version was higher. This may indicate that the level of acceptance depends not only on the amount of insect added but also on the characteristics of the product itself, which serves as the matrix.

The multiple regression analysis conducted indicated that the most important predictors shaping the final acceptance were taste and texture. These are features that should provide guidance to food producers on which features are crucial when designing this type of food. Many researchers emphasize in their studies that the addition of insect powder worsens the texture of food [90]. Osimani et al. demonstrated this by analyzing the rheology of cupcakes with the addition of *T. molitor* and *A. domesticus* compared to a control sample. Similar results were obtained by Khatun et al. [91], who partially replaced wheat flour with cricket powder in chapatti, which consequently worsened the structural properties. The dough was harder and less extensible. However, in the studies by Pauter et al. [92] muffins with added cricket were rated higher in terms of taste and texture than the control sample. Yet, the control sample received a higher overall acceptance rating than the variants with added cricket.

In our study, there were no statistically significant differences in texture between the control sample and the variants with added insects. Only muffins with added *R. differens* were rated significantly lower in both the sweet and savory versions. Perhaps the addition of insect powder was insufficient to significantly affect the rheological characteristics. Numerous studies show that adding insects to emulsions and gels can limit many structural and technological features of food products in the context of commercial use [93]. Meanwhile, Kim et al. [94] studied the impact of three insects, *T. molitor*, *A. dichotoma*, and *P. brevitarsis seulensis*, on the rheology of emulsions and found that *T. molitor* was the most suitable for use as a meat substitute in terms of its physicochemical and rheological properties. Also, Çabuk [95] suggests that enriching with powder from edible insects resulted in a denser structure and reduced specific volume compared to the control. Buns enriched with powder from locusts and mealworms showed a significantly softer consistency with lower elasticity, cohesiveness, and chewiness.

Taste is one of the most important predictors influencing the acceptance of food products, including those based on edible insects. Consumer studies indicate that the perceptions of taste and sensory quality of products are crucial for their market acceptance. In our study, taste was the deciding factor in the final acceptance of the various variants tested. Control samples without the addition of insect powder were rated highest, consistent with previous studies [96]. All other samples were rated lower; however, the taste of muffins with grasshopper flour received the lowest ratings and also the lowest overall acceptance. This confirms other researchers’ findings that *R. differens* does not present high market potential [97]. It is important to emphasize that taste in all samples was rated higher in the sweet version. This shows that sensory perception, preparation, presentation, and adaptation to consumer preferences in a particular part of the world are most crucial. In studies conducted by Hartmann and Siegrist [98], consumers showed increased willingness to eat insect-based products when they were properly prepared and presented as tasty and sensory appealing. Powdered forms of insects, such as flour, can be more easily accepted when used to enrich well-known and commonly liked products, such as cookies, buns, or bread. In our study, the addition consisted only of powdered lyophilized insects. The main aim was to compare different insects and their impact on palatability and other quality indicators, and to point out any differences in acceptance between the sweet and salty versions. This was an attempt to create a modification that would be highly rated and simultaneously have the greatest economic potential. The study was conducted on a relatively homogeneous group of subjects. Perhaps results in a different age group or in another area could vary and provide different suggestions. Many authors [65,99,100,101] mention that the characteristics of the target group may not provide a complete picture, although another study [84] did not confirm any dependence of acceptance ratings on age. Acceptance levels between seniors and young adults did not differ significantly. Due to the fact that the study participants were healthy, young, and without allergies and did not take medications on a regular basis, our results cannot be generalized to the entire population.

In addition, sensory evaluation was conducted on untrained participants. Therefore, they probably showed varying levels of sensory sensitivity. To increase the reliability of this type of research in relation to the general population, the results obtained from different age groups should be compared in one study. What will be the next step of our research?

Cooking demonstrations, tastings, and informational campaigns can effectively influence the perception of insects as tasty and healthy food options. Ultimately, understanding and addressing taste issues is key to increasing the acceptance of insect-based food products on global markets. The analysis conducted helped determine which insects have the greatest sensory and market potential and which recipes are most conducive to acceptance.

Future research should focus on other factors that determine the acceptance of specific products and their various versions. It is possible that the eating behaviors of participants had a minor impact on the results of overall acceptance and individual sensory parameters. For example, individuals who do not consume sponge cake, muffins, or generally rarely eat sweets may tend to subjectively assess sensory scores through the lens of their own beliefs or prejudices. For this reason, studies based on direct consumption by participants pose a significant challenge. However, the knowledge gained from these studies can be a valuable guide for food producers on how to design functional foods, and for dietitians and doctors on how to compose diets to address protein deficiencies. The results of our research fill a gap concerning the impact of attitudes and food neophobia on the level of consumer acceptance. The study is particularly valuable as it was conducted in real-life conditions with a large group of consumers, using a matrix of sweet and savory cookies enriched with four species of insects. This comprehensive comparative study, presented in a model-based approach, is the first of its kind reported in the literature.

## 5. Conclusions

Based on the analysis conducted in this study, it was found that consumer attitudes and the level of neophobia towards food significantly influenced the acceptance of muffins containing insect powder. Participants with lower levels of neophobia and positive attitudes towards insect consumption generally rated the insect-enhanced muffins higher compared to those with higher levels of neophobia and negative attitudes. The results indicate that overcoming psychological barriers associated with consuming insects is crucial for increasing the acceptance of insect-based food products. The study demonstrated that muffins with insect powder had lower overall acceptance compared to control samples, highlighting the market challenges associated with integrating insect ingredients into commercial offerings. Versions of the muffins made with insect powder and sweetened were rated higher, which also provides insights into dietary preferences and habits. Variants containing grasshopper powder were rated the lowest for both taste versions. Meanwhile, the greatest taste potential was seen in products based on buffalo worm powder. Multiple regression analyses indicated that taste is the decisive factor influencing the acceptance of food products containing insect powder, regardless of the insect species. Further research on the use of insects in the diets of various societies and effective marketing and educational strategies are necessary to break down cultural barriers and eating habits.

## Figures and Tables

**Figure 1 foods-13-01735-f001:**
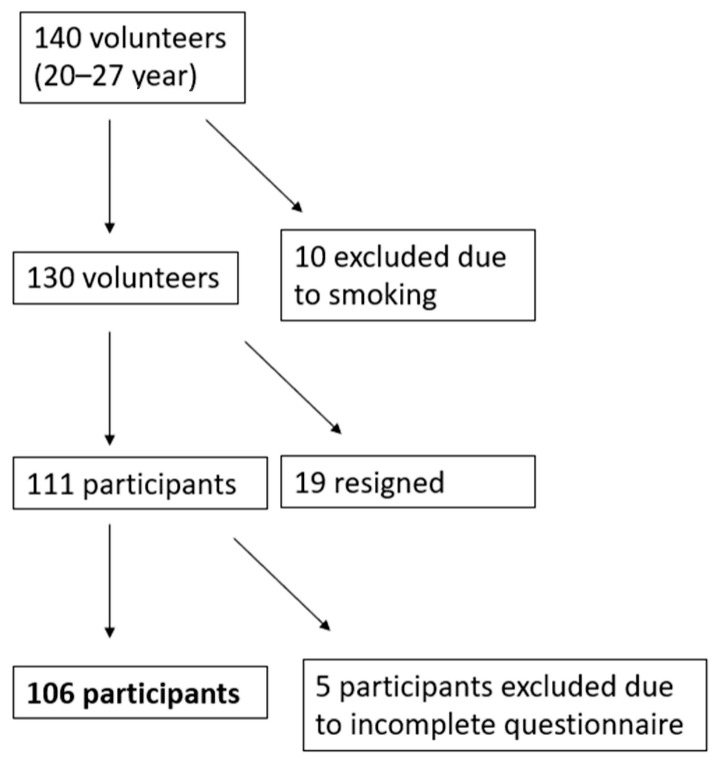
Characteristics of the study group.

**Figure 2 foods-13-01735-f002:**
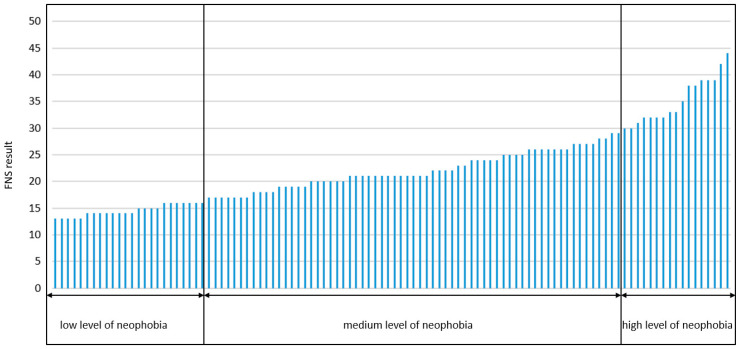
Distribution of food neophobia levels in the study group.

**Figure 3 foods-13-01735-f003:**
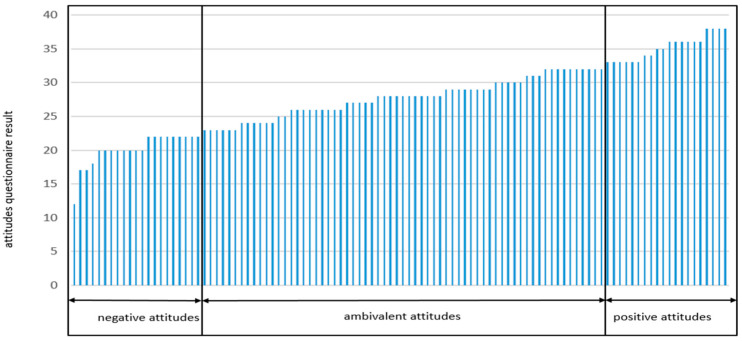
Distribution of scores obtained in the attitude questionnaire among the study group.

**Figure 4 foods-13-01735-f004:**
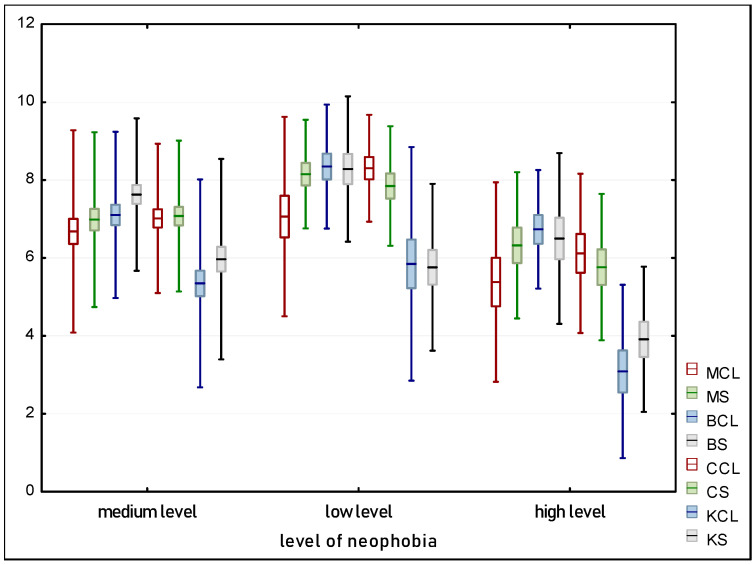
Overall acceptance relative to the level of neophobia.

**Figure 5 foods-13-01735-f005:**
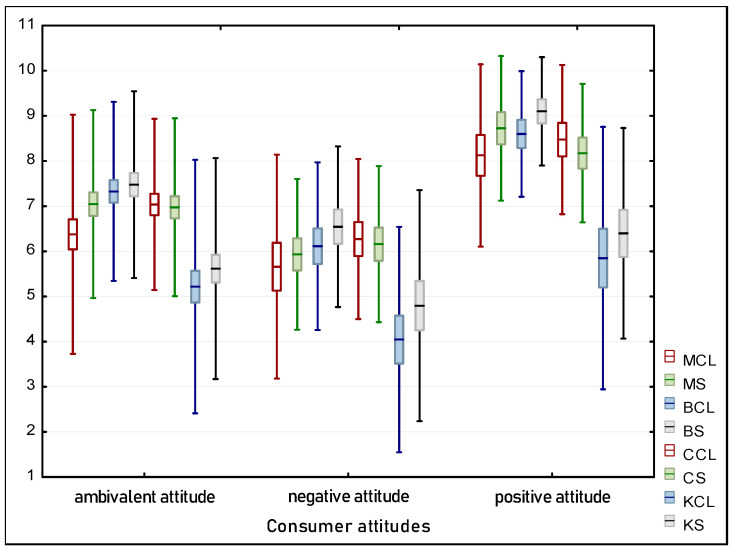
Overall Acceptance Relative to Attitudes.

**Table 1 foods-13-01735-t001:** Muffin ingredients (in grams).

Component	Control Muffin (Sweet) [g]	Control Muffin (Savory) [g]	Insect-Based Muffin (Sweet) [g]	Insect-Based Muffin (Savory) [g]
wheat flour	200	200	200	200
eggs	120	120	120	120
milk	115	115	115	115
canola oil	132	132	125	125
baking powder	10	10	10	10
sugar	18	0	18	0
salt	0	3	0	3
insect flour	0	0	35.5	35.5
wheat bran	28.5	28.5	0	0

**Table 2 foods-13-01735-t002:** Chemical composition of edible insect powder (%).

Insect Species	Protein	Ash	Fat	Insoluble Fiber Fraction Including Chitin	Soluble Fiber Fraction	Total Fiber	Moisture
*A.diaperinus*	49.53 ± 0.19 ^b^	4.71 ± 0.04 ^a^	26.44 ± 0.28 ^b^	12.96 ± 0.07 ^c^	-	12.96 ± 0.03 ^c^	3.30 ± 0.03 ^a^
*A.domesticus*	55.18 ± 0.16 ^a^	4.34 ± 0.02 ^c^	29.01 ± 0.12 ^a^	18.48 ± 0.09 ^b^	0.24 ± 0.03 ^b^	18.72 ± 0.06 ^b^	2.45 ± 0.03 ^b^
*T.molitor*	45.39 ± 0.09 ^c^	3.86 ± 0.03 ^d^	14.29 ± 0.16 ^c^	11.56 ± 0.05 ^d^	0.36 ± 0.01 ^a^	11.92 ± 0.06 ^d^	1.62 ± 0.13 ^c^
*R.differens*	42.96 ± 0.13 ^d^	4.48 ± 0.05 ^b^	9.60 ± 0.11 ^d^	19.94 ± 0.09 ^a^	-	19.94 ± 0.07 ^a^	1.73 ± 0.16 ^c^

^a–d^ The mean values followed by the same letter are not significantly different, according to *p* < 0.05.

**Table 3 foods-13-01735-t003:** Acceptability of insect-based muffins.

Sample	Type of Additive	Variant	Appearance	Aroma	Taste	Texture	Acceptance
OCL	Control	salty	9.10 ± 1.27 ^a^	8.83 ± 1.66 ^a^	8.20 ± 1.84 ^ac^	8.30 ± 1.78 ^a^	7.50 ± 1.92 ^ab^
MCL	*T.molitor*	salty	9.02 ± 1.65 ^a^	7.93 ± 2.36 ^b^	6.96 ± 2.70 ^b^	7.71 ± 2.37 ^a^	6.56 ± 2.61 ^b^
BCL	*A.diaperinus*	salty	9.02 ± 1.49 ^a^	8.76 ± 1.74 ^ab^	7.95 ± 1.99 ^ac^	7.99 ± 2.03 ^a^	7.32 ± 2.00 ^ab^
CCL	*A.domesticus*	salty	9.30 ± 1.27 ^a^	8.51 ± 1.76 ^ab^	7.59 ± 1.98 ^ab^	8.14 ± 1.94 ^a^	7.15 ± 1.95 ^ab^
GCL	*R.differens*	salty	7.82 ± 2.28 ^b^	6.77 ± 2.54 ^c^	5.55 ± 2.79 ^d^	6.76 ± 2.79 ^b^	5.09 ± 2.80 ^c^
OS	Control	sweet	8.80 ± 1.50 ^a^	8.66 ± 1.62 ^ab^	8.70 ± 1.61 ^c^	8.24 ± 1.60 ^a^	7.82 ± 1.75 ^a^
MS	*T.molitor*	sweet	8.92 ± 1.52 ^a^	8.29 ± 2.07 ^b^	7.51 ± 2.32 ^ab^	8.12 ± 1.80 ^a^	7.13 ± 2.10 ^ab^
BS	*A.diaperinus*	sweet	9.05 ± 1.59 ^a^	8.79 ± 1.82 ^a^	7.80 ± 2.27 ^abc^	7.87 ± 2.39 ^a^	7.59 ± 2.03 ^a^
CS	*A.domesticus*	sweet	8.72 ± 1.67 ^a^	8.44 ± 1.90 ^ab^	7.28 ± 2.03 ^ab^	8.02 ± 1.80 ^a^	7.03 ± 1.94 ^ab^
GS	*R.differens*	sweet	7.81 ± 2.36 ^a^	8.29 ± 1.91 ^ab^	5.60 ± 2.44 ^d^	7.52 ± 2.33 ^b^	5.55 ± 2.48 ^c^

^a–d^ The mean values followed by the same letter are not significantly different, according to *p* < 0.05.

**Table 4 foods-13-01735-t004:** Statistics of the relationship between sweet and savory muffin versions.

Insect Powder Sample	Type of Test	Statistical Value	*p*
MCL/MS	Shapiro-Wilk test	0.9689	0.0136
	Wilcoxon test		0.0215 *
BCL/BS	Shapiro-Wilk test	0.9461	0.0003
	Wilcoxon test		0.7096
CCL/CS	Shapiro-Wilk test	0.9472	0.0004
	Test Wilcoxon		0.1419
GCL/GS	Shapiro-Wilk test	0.9836	0.2155
	Wilcoxon test		0.0239 *

* *p* < 0.05.

**Table 5 foods-13-01735-t005:** Multiple equations of overall muffin acceptability.

Type of Additive	Regression Equation	R^2^
Control (CS)	y = −2.73 + 0.58x_1_ − 0.03x_2_ + 0.26x_3_ + 0.42x_4_	0.70
*T. molitor* (MS)	y = −2.18 + 0.37x_1_ − 0.10x_2_ + 0.24x_3_ + 0.55x_4_	0.75
*A. diaperinus* (BS)	y = 0.879 + 0.54x_1_ − 0.18x_2_ + 0.10x_3_ + 0.35x_4_	0.85
*A. domesticus* (CS)	y = −0.35 + 0.64x_1_ − 0.01x_2_ + 0.14x_3_ + 0.18x_4_	0.70
*R. differens* (GS)	y = − 1.70 +0.69x_1_ + 0.14x_2_ + 0.11x_3_ + 0.18x_4_	0.85
Control (CCL)	y = −2.31 + 0.40x_1_ + 0.28x_2_ + 0.21x_3_ + 0.26x_4_	0.65
*T. molitor* (MCL)	y = −1.62 + 0.6x_1_ + 0.10x_2_ + 0.15x_3_ + 0.23x_4_	0.78
*A. diaperinus* (BCL)	y = −1.25 + 0.58x_1_ − 0.11x_2_ + 0.10x_3_ + 0.26x_4_	0.72
*A. domesticus* (CCL)	y = −1.14 + 0.46x_1_ − 0.17x_2_ + 0.13x_3_ + 0.27x_4_	0.61
*R. differens* (GCL	y = −2.05 + 0.50x_1_ − 0.14x_2_ + 0.23x_3_ + 0.24x_4_	0.78

R^2^—determination coefficient, y—preference; x_1_—taste; x_2_—aroma; x_3_—appearance; x_4_—texture.

## Data Availability

The data presented in this study are available on request from the corresponding author. The data are not publicly available due to privacy restrictions.

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
