# Peer review of "Acceptance of Muffins (Sweet and Savory) with the Addition of T. molitor, A. diaperinus, A. domesticus, R. differens, Considering Psychological Factors (Food Neophobia Scale, Consumer Attitude)"

_foods, 2024, doi:10.3390/foods13111735_

Round 1
Reviewer 1 Report
Comments and Suggestions for Authors
The paper is not well structured and the literature review is absent.
I suggest to reconsider it after major revision
The paper is not well structured and the literature review is absent.
I suggest the following revisions:
Abstract:
Please explain in a better way the study's aims (lines 30-33)
Introduction:
Line 38: Rising incomes and urbanization are causing a global shift in dietary habits Please cite source
Line 40: Many societies (i.e.?)
Line 54: considering the growing population 54 and the high environmental costs of obtaining animal protein. Please report some data
Lines 60-63: Please cite sources
Please explain your research questions (RQs)
2. Literature review
Please include this section
3. Materials and Methods
Lines 104-115:The FNS questionnaire included the following statements.
Why these statements motive with a more review of the previous literature
Lines 140-148: Motive with literature review
4. Results
Please summarize findings and use a more comprensive language. Motive your findings in relation to previous literature.
5. Discussion
Please identify the gap in the literature and motive in a better way the scientific relevance of your work
Author Response
We sincerely thank you for your time and effort in preparing this review. We have taken all your valuable suggestions into account. Our responses to each of your comments are detailed below and incorporated into the manuscript text.
The paper is not well structured and the literature review is absent.
I suggest to reconsider it after major revision
Abstract:
Please explain in a better way the study's aims (lines 30-33)
The abstract has been revised. The research objective was not added in lines 30-33 because it was included at the beginning of the abstract.
Introduction:
Line 38: Rising incomes and urbanization are causing a global shift in dietary habits Please cite source
Line 40: Many societies (i.e.?)
Line 54: considering the growing population 54 and the high environmental costs of obtaining animal protein. Please report some data
Lines 60-63: Please cite sources
Please explain your research questions (RQs)
In order to address all comments from both reviewers, we have thoroughly revised the Introduction section. Where suggested, we have enriched the manuscript with new and up-to-date publications.
- Literature review
Please include this section
The manuscript is extensive and constitutes a research study; therefore, the literature review is not as detailed and must be limited in accordance with the journal's requirements.
- Materials and Methods
Lines 104-115:The FNS questionnaire included the following statements.
The FNS questionnaire was fully adopted using a widely recognized methodology, which has been noted in the manuscript text.
Why these statements motive with a more review of the previous literature
Lines 140-148: Motive with literature review
Since the questionnaire was original but based on previous similar publications, those references have been added and have enriched the methodology section.
- Results
Please summarize findings and use a more comprensive language. Motive your findings in relation to previous literature.
In this section, we have added information regarding the gap in the literature and the scientific value of our study. Once again, we greatly appreciate all your helpful remarks. We have made every effort to improve this manuscript.
In accordance with editorial requirements, the Results section does not include direct discussion; therefore, comparisons with previous publications are presented in the Discussion section. Additional references to some studies have been included in this part.
- Discussion
Please identify the gap in the literature and motive in a better way the scientific relevance of your work
In this section, we have added information regarding the gap in the literature and the scientific value of our study.
Once again, we greatly appreciate all your helpful remarks. We have made every effort to improve this manuscript.
Reviewer 2 Report
Comments and Suggestions for Authors
Dear authors,
The manuscript entitled Acceptance of muffins (sweet and savory) with the addition of T. molitor, A. diaperinus, A. domesticus, R. differens, considering psychological factors (Food Neophobia Scale, Consumer Attitude) is interesting and relevant to the aims and scope of the journal. However, you could enhance the manuscript by incorporating the following suggestions:
Statements such as "products based on buffalo worms were seen as having the greatest potential" lack specificity regarding what "greatest potential" refers to (e.g., taste, acceptance, market viability).
The abstract is lengthy and contains redundant information, which can obscure the main findings and make it difficult for readers to quickly grasp the study's key points.
There are some typographical errors such as "insect power" instead of "insect powder".
The introduction covers multiple topics without a clear and logical flow. It begins with global dietary shifts, touches on environmental impacts, moves to nutritional content, and ends with cultural acceptance. Each paragraph introduces new topics without seamlessly connecting them.
The participants are homogeneous in terms of health status, as they have no allergies, chronic diseases, or regular medication use. This limits the generalizability of the findings to populations with different health conditions. This should be specified in the limitations and suggestions for future research section. I suggest that you include limitations and suggestions for future research in the conclusion.
The statistical methods used (Shapiro-Wilk, Wilcoxon, Kruskal-Wallis, and regression analysis) are appropriate but should include corrections for multiple comparisons to avoid type I errors. Additionally, more advanced techniques, like mixed-effects models, could better handle the repeated measures and participant variability.
While targeted marketing strategies are mentioned as crucial for promoting insect-based products, the discussion does not delve deeply into specific marketing approaches or their effectiveness. More detailed insights into effective marketing tactics could enhance the practical applicability of the research.
While the importance of consumer education is highlighted, the discussion does not provide concrete strategies for effectively educating consumers about insect-based foods. Addressing this gap could facilitate broader acceptance and market penetration.
Author Response
Thank you for your review and valuable comments. Below, we will address each of your points in detail and strive to improve the content of the manuscript accordingly.
Statements such as "products based on buffalo worms were seen as having the greatest potential" lack specificity regarding what "greatest potential" refers to (e.g., taste, acceptance, market viability).
The specific type of potential has been clarified in the text. Indeed, it was a case of shorthand that could cause confusion. The necessary amendment has been made in line [29,507,680].
The abstract is lengthy and contains redundant information, which can obscure the main findings and make it difficult for readers to quickly grasp the study's key points.
The abstract has been revised in accordance with the reviewer's suggestion.
There are some typographical errors such as "insect power" instead of "insect powder".
Thank you for this valuable remark. All identified errors have been corrected accordingly.
The introduction covers multiple topics without a clear and logical flow. It begins with global dietary shifts, touches on environmental impacts, moves to nutritional content, and ends with cultural acceptance. Each paragraph introduces new topics without seamlessly connecting them.
In order to address all comments from both reviewers, we have thoroughly revised the Introduction section. Where suggested, we have enriched the manuscript with new and up-to-date publications.
The participants are homogeneous in terms of health status, as they have no allergies, chronic diseases, or regular medication use. This limits the generalizability of the findings to populations with different health conditions. This should be specified in the limitations and suggestions for future research section. I suggest that you include limitations and suggestions for future research in the conclusion.
This section has been moved to the discussion at line [635].
The statistical methods used (Shapiro-Wilk, Wilcoxon, Kruskal-Wallis, and regression analysis) are appropriate but should include corrections for multiple comparisons to avoid type I errors. Additionally, more advanced techniques, like mixed-effects models, could better handle the repeated measures and participant variability.
Thank you for your advice on statistics chosen and modelling method. Indeed it would be feasible to propose different test or extend the reasoning by new analysis. The study was conducted with significance level α=0.05, which determines maximum acceptable risk of unfounded rejecting null hypothesis - committing type I error. We are aware it is unavoidable to commit such mistakes, since there is always a chance of the results being completely random. However we opted for performing a test that would accurately reflect the subject’s characteristic and allow reliable analysis. We feel further calculations would hinder interpretation of the results.
When it comes to modelling, its aim was to examine dependency of overall preference on different aspects of the muffins. Therefore clustered linear regression was applied. Mixed effects models would indeed come in handy if interviewees’ specific effect was taken into account. However in this study it was not our goal to recognize such effect. We focused on each muffin’s type rating separately, as if it was a different study, hence same participant evaluating two different muffin types may be treated as two different people.
While targeted marketing strategies are mentioned as crucial for promoting insect-based products, the discussion does not delve deeply into specific marketing approaches or their effectiveness. More detailed insights into effective marketing tactics could enhance the practical applicability of the research.
In several places, marketing strategies were discussed. Wherever necessary, we have improved each section and added specific references at line [514-521].
While the importance of consumer education is highlighted, the discussion does not provide concrete strategies for effectively educating consumers about insect-based foods. Addressing this gap could facilitate broader acceptance and market penetration.
Thank you. This is a very valuable comment. The elements concerning marketing and education have been outlined in the discussion section and improved with an additional source.
Once again, we greatly appreciate all your helpful remarks. We have made every effort to improve this manuscript.
Round 2
Reviewer 1 Report
Comments and Suggestions for Authors
The authors have notable improved their work and now it is suitable for the publication